# Socioeconomic impact of the COVID-19 crisis and early perceptions of COVID-19 vaccines among immigrant and nonimmigrant people living with HIV followed up in public hospitals in Seine-Saint-Denis, France

**Pauline Penot**[1,2]*, **Julie Chateauneuf**[1,3], **Isabelle Auperin**[4], **Hugues Cordel**[5], **Valerie-Anne Letembet**[1], **Julie Bottero**[5], **Johann Cailhol**[5]

1 CEGIDD (Sexual Health and Infectious Disease Clinic), Centre Hospitalier André Grégoire, Montreuil, France, 2 Centre Population et Développement (Ceped), Institut de Recherche pour le Développement (IRD) et Université de Paris, Paris, France, 3 Perinatal Network "Naître dans l'Est Francilien", Centre Hospitalier André Grégoire, Montreuil, France, 4 Addictology and Infectious Disease Unit, Centre Hospitalier André Grégoire, Montreuil, France, 5 Infectious and Tropical Disease Ward, Avicenne Hospital, Bobigny, France

* pauline.penot@ght-gpne.fr

**Data Availability Statement:** All data underlying the reported results are provided in the submitted

## Abstract

The burden of the first year of the coronavirus disease 2019 (COVID-19) pandemic was greater for vulnerable populations, such as immigrants, people living in disadvantaged urban areas, and people with chronic illnesses whose usual follow-up may have been disrupted. Immigrants receiving care for HIV in Seine-Saint-Denis' hospitals have a combination of such vulnerabilities, while nonimmigrant people living with HIV (PLWHIV) have more heterogeneous vulnerability profiles. The ICOVIH study aimed to compare the socioeconomic effects of the COVID-19 crisis as well as attitudes toward COVID-19 vaccination among immigrant and nonimmigrant PLWHIV. A questionnaire assessed vulnerabilities prior to the COVID-19 epidemic and the impact of the early epidemic on administrative, residential, professional, and financial fields. We surveyed 296 adults living with HIV at four hospitals in Seine-Saint-Denis, the poorest metropolitan French department, between January and May 2021. Administrative barriers affected 9% of French-born *versus* 26.3% of immigrant participants. Immigrants experienced financial insecurity and hunger more often than nonimmigrant participants (21.8% *versus* 7.1% and 6.6% *versus* 3%, respectively). Spontaneous acceptance of vaccination was higher among nonimmigrant than among immigrant participants (56.7% *versus* 32.1%), while immigrants were more likely to wait for their doctor's recommendation or for their doctor to convince them than their French-born counterparts (34.2% *versus* 19.6%). The trust-based doctor–patient relationship established through HIV follow-up appeared to be a determining factor in the high acceptance of the COVID-19 vaccine among immigrant participants.

article and in the supporting information. However, the individual data of participants cannot be shared due to ethical and legal restrictions. The participants are a vulnerable population and the data contain sensitive information about their administrative status, experience of violence, sexuality and health, which can be used to re-identify them. Due to privacy agreements and the nature of our data, ethics committees and the French data protection authority do not allow the data to be made available to the public. All relevant data can be requested from the principal investigator of the study: pauline.penot@ght-gpne.fr or from the hospital carrying out the project, writing to ag.cegidd@ght-gpne.fr.

**Funding:** The authors received no personal funding for this work. The French AIDS society (Société Française de Lutte contre le Sida) provided trained interviewers' time to this study.

**Competing interests:** The authors have declared that no competing interests exist.

## Introduction

Individuals who are born foreigners abroad and reside in France are defined by the French High Council for Integration as immigrants: in France, as elsewhere in Europe [1–3], immigrants were the most hard hit by coronavirus disease 2019 (COVID-19). During the first wave, both seroprevalence and mortality data confirmed a greater COVID-19-related risk among immigrants: COVID-19 seroprevalence was 9.4% among people born outside Europe compared to 4.1% in the overall French population [4]. The excess overall mortality rate was 114% among people born in sub-Saharan Africa compared to 22% among people born in France in March and April 2020 [5]. Moreover, since they were more likely than nonimmigrants to be in a disadvantaged situation prior to the epidemic, immigrants suffered more from the social and psychological consequences of the crisis [6, 7].

Seine-Saint-Denis, located in the northeastern suburbs of Paris, is the department with the highest rate of immigration and the highest rate of poverty in mainland France [8]. Therefore, the consequences of the COVID-19 crisis have been considerable in this department: the first epidemic wave led to an excess overall mortality rate of 124% between 1 March and 30 April 2020 compared to the same period in 2019. This excess rate was higher than that in Paris despite the Seine-Saint-Denis population being younger [9]. Immigrants, especially those from sub-Saharan Africa, are more affected by HIV and migration, and COVID-19 and HIV might well constitute a new syndemic [10, 11].

Concerns have also been raised about people living with HIV (PLWHIV) not being receiving timely and equitable access to health care [12, 13]. However, to date, little is known about the specific social impacts of the COVID-19 pandemic on PLWHIV, especially those with social vulnerabilities [14, 15]. In our respective HIV clinics, we observed an increasing number of immigrant patients who had lost their employment, experienced overcrowding or loss of housing and fallen into poverty, as reported elsewhere [16, 17].

Attitudes of PLWHIV toward COVID-19 vaccination are also poorly documented and inconsistent: older PLWHIV were more likely to report an intention to vaccinate in studies conducted in the USA [18], Nigeria [19] and Canada [20], but not in a cohort of men having sex with men (MSM) from China, in which concern about disclosure of HIV status was one of the top reasons not to initiate COVID-19 vaccination [21], nor in an Indian cohort, in which lack of confidence in common sources of vaccine-related information, including doctors, emerged as the key parameter in vaccine reluctance [22]. French immigrants have been shown to be more reluctant to receive the COVID-19 vaccine (first-generation African/Asian immigrants OR = 1.16 (95% CI: 1.04–1.30)) [23]. Thus, to our knowledge, vaccine hesitancy has not yet been compared between immigrant and nonimmigrant PLWHIV.

The ICOVIH study aimed to compare the socioeconomic effects of the COVID-19 crisis as well as attitudes toward COVID-19 newly marketed vaccines among immigrant and nonimmigrant PLWHIV.

## Methods

We established a questionnaire that mainly covered the domains of essential needs (housing, employment, income, administrative status for immigrants). From January 5 to June 1, 2021, the ICOVIH study ("Impact de la crise COVID-19 sur les personnes vivant avec le VIH") was conducted among PLWHIV followed up at four public hospitals in Seine-Saint-Denis (André Grégoire Hospital in Montreuil, Avicenne Hospital in Bobigny, Jean Verdier Hospital in Bondy, Delafontaine Hospital in Saint-Denis). Eligible patients were over 18 years old and on antiretroviral therapy before the COVID-19-related first lockdown in France in March 2020. The sample was calculated on the hypothesis that immigrants living with HIV were twice as

impacted financially by the COVID-19 crisis compared to nonimmigrants living with HIV and that 20% of nonimmigrants experienced a financial impact. The sample size needed was 214. Medical doctors in charge of patients were asked to propose the survey to their patients at each planned medical visit during the study period. Upon patient acceptance, the survey was administered on the spot and face-to-face or later over the phone, depending on the patient's choice. Surveys were conducted by trained interviewers. Data were collected anonymously in the Sphinx©IQ2 database (Le Sphinx, Chavanod, France). Information collected consisted of demographic, administrative situation, COVID-19 administrative and financial impact, employment status, housing difficulty, and acceptability of COVID-19 vaccine variables (the variables are presented in detail in S1 Appendix in English and S2 Appendix in French). Immigrant and nonimmigrant participant data were compared using chi-squared or Fisher's exact tests for categorical variables and the Mann–Whitney U test for continuous variables. All analyses were performed using STATA©12.2 (StataCorp, College Station, TX, USA) software.

## Ethical considerations

Medical doctors informed patients about the study objective during their routine follow-up visits. Clarity was sought about participants' full understanding of the survey and their freedom to consent or not. Patients were aware that their refusal to participate would not affect their quality of care and that they were free to not answer all questions and to withdraw from the survey at any time.

The French Data Protection Authority (CNIL registration number 118512–1610969239) and the André Grégoire Hospital ethics committee (IRB10022022) both approved this project.

## Results

### Sociodemographic characteristics of immigrant and nonimmigrant participants

The full active file included 1735 PLWHIV. Of these, 1206 were seen at least once during the study period, among whom 380 were offered to participate and 298 accepted (25%). Because of inconsistent data that could not be corrected due to the anonymous data collection process, two participants were subsequently excluded from the analysis. Finally, 296 patients were included in the analysis, of whom 197 were born abroad and 99 were born in France.

When compared to the characteristics of the entire cohort of the participating hospitals, male participants of the ICOVIH study were less likely to be born abroad than those from the overall cohort (50.6% *versus* 60.9% for the overall PLWHIV cohort). Conversely, no difference was observed in terms of age or geographical origin between women in the ICOVIH study and women from the overall cohort.

Most immigrants were from Western Africa (42.7%) and Central Africa (32.0%). Among immigrant participants, 38.1% had a 10-year residence permit, while 5.1% remained undocumented. Most of the immigrants (81.7%) had been in France for more than 7 years (median length of stay 18 years, interquartile range 9–25 years; Table 1). Only four participants had been in France for less than 2 years.

Compared to nonimmigrants, immigrant participants were significantly more likely (i) to be women (56.8% *versus* 16.2%, $p < 0.001$) and (ii) to be younger (median age 48 *versus* 56 years, $p < 0.001$). Only one woman surveyed was pregnant during the first year of the pandemic, and she was an immigrant. MSM were far more represented among men born in France than among those born abroad (68.3% *versus* 15.7%). Household composition was significantly different in the two groups: immigrant participants were less likely to live alone

**Table 1. Characteristics of immigrant PLWHIV compared to those born in France, 2021, Seine-Saint-Denis, France.**

| | Immigrants (n = 197) | | Individuals born in France (n = 99) | | P |
|---|---|---|---|---|---|
| Sex | | | | | |
| Women | 112 | (56.8%) | 16 | (16.2%) | |
| Men | 85 | (43.2%) | 83 | (83.8%) | < **0.001** |
| Age (years) | | | | | |
| Overall | 48 | [41–56] | 56 | [49–61] | < **0.001** |
| Women | 47 | [40–55] | 54 | [47–60] | **0.010** |
| Men | 48 | [43–57] | 56 | [49–61] | < **0.001** |
| Men's sexual orientation | | | | | |
| MSM | 13 | (15.7%) | 56 | (68.3%) | < **0.001** |
| Heterosexual men | 70 | (84.3%) | 26 | (31.7%) | |
| Place of birth | | | | | |
| France (metropolitan) | NA | | 88 | (88.9%) | |
| France (overseas) | NA | | 11 | (11.1%) | |
| Europe (outside France) | 8 | (4.1%) | | | |
| Western Africa | 84 | (42.7%) | | | |
| Central Africa | 63 | (32.0%) | | | |
| Eastern Africa | 3 | (1.5%) | | | |
| North Africa | 14 | (7.1%) | | | |
| Non-French Caribbean | 13 | (6.6%) | | | |
| North America | 1 | (0.5%) | | | |
| South America | 4 | (2.0%) | | | |
| Asia | 5 | (2.5%) | | | |
| Length of stay in France** (years) | 18 | [9–25] | | NA | |
| < 7 years | 34 | (18.3%) | | NA | |
| > = 7 years | 152 | (81.7%) | | NA | |
| Administrative status | | | | | |
| French citizenship | 44 | (22.3%) | 99 | (100%) | |
| European Union or UK citizenship | 7 | (3.6%) | | | |
| 10-year residence card | 75 | (38.1%) | | | |
| 1-to-5-year residence permit | 44 | (22.3%) | | | |
| Short residence permit (<1 year) | 17 | (8.6%) | | | |
| Undocumented | 10 | (5.1%) | | | |
| COVID-19 administrative impact | | | | | |
| None | 145 | (73.7%) | 90 | (91%) | **0.001** |
| Delay in residence permit | 31 | (15.7%) | NA | NA | |
| Delay in family reunification procedure | 3 | (1.5%) | NA | NA | |
| Delay in application for a social welfare | 11 | (5.6%) | 7 | (7.1%) | |
| Delay in naturalization-civil status document | 5 | (2.5%) | 1 | (1%) | |
| Delay in professional training or diploma | 2 | (1.0%) | 1 | (1%) | |
| Current housing difficulties | | | | | |
| None | 138 | (70.1%) | 92 | (92.9%) | <**0.001** |
| Loss of housing since March 2020 | 3 | (1.5%) | 0 | (0%) | |
| Late payment of rent | 38 | (19.3%) | 0 | (0%) | |
| Overcrowding due to loss of housing by others | 3 | (1.5%) | 0 | (0%) | |
| Delay in housing application due to COVID-19 | 10 | (5.1%) | 7 | (7.1%) | |
| > = 1night in the street since March 2020 | 5 | (2.5%) | 0 | (0%) | |

*(Continued)*

**Table 1.** (Continued)

| | Immigrants (n = 197) | | Individuals born in France (n = 99) | | P |
|---|---|---|---|---|---|
| Household: Who do you live with? | | | | | |
| No one | 50 | (25.4%) | 42 | (42.4%) | <0.001 |
| Roommates/co-residents/variable | 2522 | (12.7%) | 434 | (4.0%) | |
| Partner only | 22 | (11.2%) | 34 | (34.4%) | |
| Children +/- partner | 85 | (43.1%) | 7 | (7.1%) | |
| Other family member | 15 | (7.6%) | 12 | (12.1%) | |
| Employment status | | | | | |
| Stable contract or activity | 75 | 38.1%) | 42 | (42.4%) | <0.001 |
| Unstable job, student, crisis-related reduction of working hours | 64 | (32.4%) | 111 | (11.1%) | |
| Job loss since March 20 | 8 | (4.1%) | 8 | (8.1%) | |
| Unemployed since before the crisis | 37 | (18.8%) | 10 | (10.1%) | |
| Retired, disabled or on long illness leave | 13 | (6.6%) | 28 | (28.3%) | |
| COVID-19-related financial impact | | | | | |
| No impact | 93 | (47.2%) | 59 | (59.6%) | <0.001 |
| Improved financial situation | 9 | (4.6%) | 15 | (15.15%) | |
| Loss of income without insecurity | 39 | (19.8%) | 15 | (15.15%) | |
| Loss of income leading to financial insecurity | 43 | (21.8%) | 7 | (7.1%) | |
| Loss of income leading to hunger | 13 | (6.6%) | 3 | (3.0%) | |

Data are presented as n (%) or medians [interquartile ranges]

**11 missing data points; NA: not applicable

(25.4% *versus* 42.4%) or only with their spouse (11.2% *versus* 34.4%) than their French-born counterparts. Conversely, immigrant participants lived more often with their partner and children (21.8% *versus* 4.1%) or with their children only (21.3% *versus* 3.0%). Unemployment prior to the pandemic and unstable jobs (short contract or temporary work) were more frequent in the immigrant group. Being younger, immigrant participants were less likely to be retired (3.5% *versus* 18.2%), to be disabled or to be on long sickness leave (3.1% *versus* 10.1%). Immigrants were significantly more likely to report current housing difficulties than nonimmigrants (29.9% *versus* 7.1%). A delay in paying rent was reported by 19.3% of immigrant participants and by no participant born in France.

Data not presented here were also collected on the psychological impact of COVID in these two populations of PLHIV, and on changes in the consumption of drugs during the first wave. To summarize, feelings of discouragement and expression of suicidal ideation were equally distributed between immigrant and nonimmigrant participants, while increasing, maintaining or withdrawing from drug consumption was a question almost exclusively relevant to participants born in France, as only 4 out of 196 immigrant participants declared a history of drug use.

## Impact of the COVID-19 pandemic on both groups

The administrative, financial and housing burdens related to COVID-19 were far heavier among immigrant than among nonimmigrant participants. Administrative barriers affected 9% of French-born participants *versus* 26.3% of immigrants. For the latter group, administrative issues were mostly related to their foreign status (e.g., delay in obtaining or renewing a residence permit (15.7%)), whereas HIV-positive status was supposed to entitle the holder to legal

residency in France for most countries of origin. In contrast, pandemic-related processing delays of a social housing application affected both groups (5.6% of immigrants *versus* 7.1% of nonimmigrants, Table 1).

COVID-19-related financial deprivation disproportionally affected immigrant participants, who experienced financial insecurity or even hunger more often than nonimmigrant participants (21.8% *versus* 7.1% and 6.6% *versus* 3%, respectively). Conversely, 15.1% of French-born participants declared an improvement in their financial situation, either because their incomes increased or because their expenses decreased, while such improvement was only reported by 4.6% of immigrant participants.

A COVID-19 crisis-related reduction in working hours was significantly more frequent among immigrant than among nonimmigrant participants (8.1% *versus* 2.0%). The small size of this category led us to group it together with unstable jobs, studies and training to compare similar employment situations among immigrant and French-born participants (Table 1). Job loss since the beginning of the crisis was more common among French-born than among immigrant participants (8.0% *versus* 4.1%, *p* < 0.001 for overall differences in employment).

Immigrant participants experienced loss of housing (1.5%), overcrowding due to loss of housing by others (1.5%) and nights spent in the street (2.5%), while no French-born participant reported any of those situations during the COVID-19 crisis period.

## COVID-19 vaccine acceptance among immigrant and nonimmigrant participants

Although overall acceptance of the COVID-19 vaccine was not significantly higher in the French-born group (73.3% *versus* 66.3%; Table 2), the conditions for adherence and reasons for refusal were substantially different. Spontaneous acceptance was higher among nonimmigrants (56.7% *versus* 32.1% among immigrant participants), while immigrants were far more

**Table 2. Acceptability of the COVID-19 vaccine among immigrant participants compared to those born in France, 2021, Seine-Saint-Denis, France.**

| | Immigrants (n = 184*) | | Individuals born in France (n = 97*) | | *P* |
|---|---|---|---|---|---|
| Overall acceptability | | | | | |
| Yes | 122 | (66.3%) | 74 | (76.3%) | |
| No | 62 | (33.7%) | 23 | (23.7%) | 0.083 |
| Acceptance | | | | | |
| Spontaneously | 59 | (32.1%) | 55 | (56.7%) | |
| On medical recommendation | 63 | (34.2%) | 19 | (19.6%) | **0.001** |
| No because of fear of the vaccine | 57 | (31.0%) | 20 | (20.6%) | |
| No because of low self-perception of risk | 5 | (2.7%) | 3 | (3.1%) | |
| Detailed conditions for acceptance | | | | | |
| Vaccination already started | 6 | (3.3%) | 6 | (6.2%) | |
| Immediately, without hesitation | 53 | (28.8%) | 49 | (50.5%) | |
| Only if my doctor recommends it | 25 | (13.6%) | 10 | (10.3%) | |
| Only if my doctor convinces me by answering all my questions and explaining why I should receive the vaccine | 38 | (20.6%) | 9 | (9.3%) | |
| No, I am afraid of the side effects because of my HIV | 9 | (4.9%) | 1 | (1.0%) | **0.001** |
| No, I am afraid of the side effects, but it has nothing to do with my HIV | 12 | (6.5%) | 0 | (0%) | |
| No, because I adhere to barrier measures very well and I believe that they are sufficient | 5 | (2.7%) | 3 | (3.1%) | |
| I will consider vaccination when more information or other vaccine options are available | 36 | (19.6%) | 19 | (19.6%) | |

Data are presented as n (%) or medians [interquartile ranges]; *Data were collected from the 16th participant.

likely to wait for their doctor's recommendation or for their doctor to convince them (34.2% *versus* 19.6% among French-born participants). Immigrants were also more anxious about side effects, either related (4.9% *versus* 1.0%) or unrelated (6.5% *versus* 0) to their HIV status than nonimmigrants. In both groups, the proportion of participants who postponed their potential vaccination was similar, but again, the motivations were different: three participants specified that they were waiting for a traditional vaccine technology (protein vaccine). All three were MSM born in France, while the 36 immigrants and 16 nonimmigrants remaining were not confident and waited for more information on vaccines. Self-perception of COVID-19 risk was high in both groups, with only 2.7% of immigrant and 3.1% of nonimmigrant participants considering their good compliance with barrier measures to be sufficient.

### Factors associated with COVID-19 vaccine acceptance

Spontaneous vaccine uptake, when compared with vaccine hesitancy (refusal or depending on physician's recommendations or explanations), was significantly more common among MSM (univariate regression odds ratio (OR) 3.85, confidence interval (CI) [2.04–7.25]), even after adjustment for other characteristics, including age and place of birth (multivariate adjusted OR (aOR) 2.53 [1.17–5.44]). Spontaneous acceptance was also associated with age, with PLWHIV over 60 being more likely to seek vaccination spontaneously (multivariate aOR 2.23 [1.19–4.18]).

In univariate analysis, immigrants were less likely to seek vaccination spontaneously (OR 0.36 [0.22–0.60]), and this association might have been in part related to their younger age, as it did not remain significant after adjustment for other characteristics (aOR 0.58 [0.31–1.08]). A history of COVID-19 infection also tended to be associated with spontaneous vaccine uptake after adjustment for age.

A multinomial regression model was run to compare both vaccine refusal and vaccine adherence subject to a doctor's advice with spontaneous recourse to vaccination (reference category). Immigrants were more likely to accept vaccination on medical advice than to seek it spontaneously (aOR 2.41 CI [1.13–5.17], Table 3). Conversely, immigrants did not have a higher rate of vaccine refusal. Women tended to rely more on medical recommendations than on information obtained by themselves in deciding to get vaccinated, although the small sample size did not allow us to show a significant sex difference between spontaneous and doctor-mediated vaccination uptake. Older participants were less likely to refuse vaccination than to seek it spontaneously (OR 0.37 CI [0.16–0.87]). Finally, participants who had had early COVID-19 were more likely to seek vaccination spontaneously than on medical recommendation (OR 0.37 CI [0.14–0.97]).

## Discussion

### Socioeconomic impacts of COVID-19 on immigrant and nonimmigrant PLWHIV

Immigrants from the ICOVIH study experienced more negative effects from the COVID-19 crisis than nonimmigrant participants. These effects were multidimensional: deterioration of housing conditions, decrease in income and hunger. In addition, immigrant participants were affected by administrative issues related to their foreign status, which, by definition, French-born individuals did not encounter. We therefore found among PLWHIV, as in other groups, the deepening of inequalities with the crisis [7], which was also shown in Canada [11]. It is concerning that while most immigrants had been settled in France for more than 7 years, which is considered the necessary time to obtain proper housing, employment and residency

**Table 3. Patient characteristics by acceptability of the COVID-19 vaccine, Seine-Saint-Denis, 2021 (multinomial model).**

| | Spontaneous vaccine uptake (n = 114) | Acceptability conditional upon medical recommendation/explanation (n = 82) | Refusal (n = 85) |
|---|---|---|---|
| | OR | OR [95% CI] | OR [95% CI] |
| Place of birth | | | |
| France | Ref. | Ref. | Ref. |
| Abroad | Ref | **2.41 [1.13–5.17]** | 1.80 [0.87–1.71] |
| Sex | | | |
| Men | Ref. | Ref. | Ref. |
| Women | Ref. | 1.94 [1.00–3.77] | 1.70 [0.89–3.27] |
| Age (years) | | | |
| < 60 | Ref. | Ref. | Ref. |
| > = 60 | Ref. | 0.60 [0.26–1.35] | **0.37 [0.16–0.87]** |
| Year of HIV diagnosis | | | |
| 1996 or before | Ref. | Ref. | 1.00 |
| After 1996 | Ref. | 0.50 [0.22–1.12] | 0.68 [0.30–1.51] |
| COVID-19 infection prior to survey | | | |
| No or unknown | Ref. | Ref. | Ref. |
| Yes (confirmed or suspected) | Ref. | **0.37 [0.14–0.97]** | 0.65 [0.28–1.53] |
| Employment situation | | | |
| Unemployed, retired or student | Ref. | Ref. | Ref. |
| Employed | Ref. | 1.43 [0.73–2.81] | 1.19 [0.62–2.27] |
| Housing | | | |
| Personal accommodation | Ref. | Ref. | Ref. |
| Hosted or homeless | Ref. | 1.31 [0.65–2.63] | 1.55 [0.78–3.08] |

permits [24], they were so hard hit by the crisis, as shown in our study. This means that even when immigrants are considered to be settled down, they are in a less stable situation than nonimmigrants.

In line with epidemiological data on PLWHIV in France, in the Seine-Saint-Denis department, the characteristics of PLWHIV groups differed according to their country of origin: PLWHIV born in France were more often MSM, whereas immigrant PLWHIV were mostly women born in sub-Saharan Africa and heterosexual men living in more difficult socioeconomic conditions [25, 26]. The characteristics of ICOVIH participants were similar to those of the overall cohort of PLWHIV in the participating hospitals regarding the sex proportion. However, the distribution of geographical origins was different among male participants who were less likely to be born abroad. This difference reflects the overrepresentation of French-born MSM among the patients followed by one of the medical doctors.

Immigrant PLWHIV in our sample were younger than French-born PLWHIV. This partly explains differences in household composition, with immigrants living more frequently with children than nonimmigrants. We could hypothesize that among previously officially unemployed participants, many relied on informal employment. However, during the lockdown,

outdoor circulation was authorized only with official documentation such as employer letters, and informal employment was halted. Short-term contracts were also the first contracts to not be renewed. These latter points explain why more immigrants lost income [27]. Moreover, with immigrants constantly having children at home during the lockdown (and not in schools where children could at least have lunch), their food expenditures increased and sometimes led to hunger [28]. Housing density was also one of the factors that contributed to higher COVID-19 exposure, and our sample truly reflected the social vulnerability of this population to an airborne epidemic.

At the Avicenne and Jean Verdier hospitals, a crisis support team was set up as soon as the lockdown started. Members of this team contacted PLWHIV in their cohorts by phone if they met certain vulnerability criteria. They proposed various types of support: COVID-19 advice, orientation toward nonclosed associations, treatment counseling, and food vouchers [29]. Ikambere, an association based in Seine-Saint-Denis, offers comprehensive support to HIV-positive immigrant women to break the isolation caused by a combination of HIV and precariousness. During the first year of the COVID-19 pandemic, the association faced a dramatic increase in requests related to food aid, housing insecurity, and the breaking of precarious work contracts or rights by its beneficiaries. Thanks to the financial support of private partners, Ikambere increased its distribution of food packs, released emergency financial aid and provided refuge for women who left or had to leave their accommodations. The association then set up an information campaign on vaccination and accompanied beneficiaries to vaccination centers [30]. The results of the ICOVIH survey led to strengthened ties between Montreuil Hospital and Ikambere: an office is now set up one day a week next to the medical consultation rooms to enable HIV-positive immigrant women to meet with a mediator from the association after seeing their doctor.

## Early vaccine acceptability

COVID-19 vaccine acceptability was slightly higher among French-born participants than among immigrant participants, but the difference was not significant. Surveys on vaccine acceptance usually show higher vaccine hesitancy among immigrant and minority groups than among nonimmigrants [23, 31]. Our results showed that the situation was different among PLWHIV, with high vaccine acceptance in both French-born and immigrant groups. The conditions under which immigrant participants expressed favorable attitudes toward vaccination (if the doctor recommended vaccination or if the doctor convinced them by answering all their questions and explaining why they should be vaccinated) suggest that the relationship of trust with the doctor who cares for a chronic illness was a determining factor in vaccination acceptance in this immigrant population. However, self-perception of the risk of COVID-19 in this HIV-infected population probably favored the acceptance of the doctor's vaccination recommendation, as shown by the low percentage of participants who considered that good compliance with barrier measures would be sufficient protection against COVID-19. Participants who had had an early COVID-19 infection were more likely to be vaccinated spontaneously than by medical recommendation, as they probably relied on their own experience to make this choice. Older participants were more likely to seek vaccination spontaneously than to refuse it, probably because of a higher self-perceived risk of developing a severe form of COVID-19. Surprisingly, a history of COVID-19 infection also tended to be associated with spontaneous vaccine uptake after adjustment for age (but not for comorbidities, which were not surveyed). This finding contradicts the idea that a patient confident of being immune would be less inclined to be vaccinated. It suggests, instead, that PLWHIV who had COVID-19 during the first year of the pandemic were generally symptomatic and afraid of becoming ill again.

In the general French population, male sex, high socioeconomic status and older age were positively associated with adherence to COVID-19 vaccination [32, 33].

In a study carried out in a cohort of PLWHIV in the Paris region, Vallée et al. estimated COVID-19 vaccine hesitancy to be 29%. No information was available on the participants' place of birth or socioeconomic conditions. Female sex was associated with vaccine hesitancy in univariate analysis, but sex remained nonpredictive in multivariate analysis, potentially due to the low representation of women in the sample (23%). Age was not associated with COVID-19 vaccine hesitancy. To the statement "I trust the information I receive about the COVID-19 vaccine from my doctor(s)", 97% of the PLWHIV in the vaccine acceptance group and 75% in the vaccine hesitancy group answered that they did, a significant difference (p < 0.001), making trust in the doctor's advice a key factor in adherence to vaccination [34].

Health care providers were identified as the most trusted advisors and influencers of vaccination decisions [35]. The central role played by physicians for immigrants with chronic illnesses has been demonstrated in a previous large study involving PLWHIV from Seine-Saint-Denis and neighboring districts [36]. Trust in the health system and in vaccines is possible for immigrants and people living in precarious situations: we can imagine that here, the relationship built up over time with a doctor and a team enables most immigrant PLWHIV to overcome their concerns about a new vaccine, whereas immigrants who are not engaged in care miss this opportunity.

## Strengths and limitations

We conducted a cross-sectional, single-department study on a small sample of PLWHIV. The heavy workload faced by infectiologists at the beginning of 2021 made it impossible to systematize the proposal to all PLWHIV seen in consultations over the period. In addition, the anonymous data collection process chosen for this study did not allow us to review incomplete observations, forcing us to exclude data from participants for whom inconsistencies or too much missing data were identified retrospectively. Finally, comorbidities and educational level were not surveyed, whereas such data would have refined the analysis of determinants of vaccine acceptance.

## Conclusion

The ICOVIH survey showed the cumulative and multidimensional vulnerability faced by immigrant PLWHIV, who were likely to have already been in disadvantaged situations compared to nonimmigrant PLWHIV, during the first lockdown. This study also showed that HIV-infected immigrants had confidence in the recommendation to vaccinate coming from their physician and in their physician's explanations: this confidence enabled them to catch up with a vaccination intention rate close to that of PLWHIV born in France. The physician is a privileged source of information, a focal point for people who have a combination of a chronic illness and social vulnerability. For the most vulnerable, the doctor–patient relationship is crucial, particularly for the implementation of preventive measures. This relationship will be regularly mobilized as viruses emerge and re-emerge and as new mRNA vaccines are developed in response to such emergencies.

## Supporting information

**S1 Appendix. Impact of the covid crisis on PLWHIV.**
(DOCX)

**S2 Appendix. Impact de la crise covid sur les PvVIH.**
(DOC)

## Acknowledgments

The authors thank Agnès Viot, Mélanie Billi, Annabel Desgrées du Loû, Dr Pascal Pugliese, Dr Anaenza Maresca, Dr Frederic Méchaï, Dr Marie Poupard, Dr Nolan Hassold-Rugolino, Dr Julie Figoni, Dr Claire Tantet, Audrey Guerizec, Gwen Hamet, Guy Nielsen and Karna Coulibaly.

## Author Contributions

**Conceptualization:** Pauline Penot, Julie Chateauneuf, Hugues Cordel, Johann Cailhol.

**Data curation:** Pauline Penot, Julie Chateauneuf.

**Formal analysis:** Pauline Penot, Hugues Cordel.

**Investigation:** Pauline Penot, Julie Chateauneuf, Isabelle Auperin, Hugues Cordel, Valerie-Anne Letembet, Julie Bottero.

**Methodology:** Pauline Penot, Hugues Cordel, Johann Cailhol.

**Project administration:** Pauline Penot.

**Software:** Pauline Penot, Julie Chateauneuf.

**Supervision:** Pauline Penot, Isabelle Auperin, Valerie-Anne Letembet, Julie Bottero, Johann Cailhol.

**Validation:** Pauline Penot, Isabelle Auperin, Valerie-Anne Letembet.

**Writing – original draft:** Pauline Penot, Johann Cailhol.

**Writing – review & editing:** Pauline Penot, Julie Chateauneuf, Hugues Cordel, Julie Bottero, Johann Cailhol.

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
