## [Decision Letter · Decision Letter 0]

10 Apr 2023

PONE-D-22-26796Socio-economic impacts and Covid19 vaccine perception among migrant and non-migrant PLWHIV in Seine Saint Denis, FrancePLOS ONE

Dear Dr. Penot,

Thank you for submitting your manuscript to PLOS ONE. After careful consideration, we feel that it has merit but does not fully meet PLOS ONE’s publication criteria as it currently stands. Therefore, we invite you to submit a revised version of the manuscript that addresses the points raised during the review process.

We look forward to receiving your revised manuscript.

Kind regards,

Ali B. Mahmoud, Ph.D.

Academic Editor

PLOS ONE

Journal Requirements:

3. Please amend the manuscript submission data (via Edit Submission) to include author Isabelle AUPERIN, Valerie-Anne LETEMBET.

4. Please amend your authorship list in your manuscript file to include author CEGIDD hospitalier de Montreuil

Reviewers' comments:

Reviewer's Responses to Questions

**Comments to the Author**

1. Is the manuscript technically sound, and do the data support the conclusions?

Reviewer #1: Partly

Reviewer #2: Yes

Reviewer #3: Partly

2. Has the statistical analysis been performed appropriately and rigorously? 

Reviewer #1: I Don't Know

Reviewer #2: No

Reviewer #3: No

3. Have the authors made all data underlying the findings in their manuscript fully available?

Reviewer #1: Yes

Reviewer #2: Yes

Reviewer #3: Yes

4. Is the manuscript presented in an intelligible fashion and written in standard English?

Reviewer #1: No

Reviewer #2: Yes

Reviewer #3: No

5. Review Comments to the Author

Reviewer #1: The ICOVIH study compared the socio-economic effects of COVID-19 crisis and attitudes towards COVID-19 vaccination, between migrant and non-migrant PLWHIV in Seine-Saint Denis. As expected, migrants had more social vulnerability than non-migrants. A trust-based doctor-patient relationship established through HIV follow-up appeared as a determining factor in acceptance of COVID-19 vaccination among migrants.

I have a few questions :

1. How did the questionnaire take into account comprehension, language ?

2. How many woman were pregnant and was the proportion different in between groups ? pregnancy or planned pregnancy have been shown to

3. What was the proportion of drug use in the two groups ?

4. How many patients with a history of covid in each group ?

5. Was there any information on the background of the French-born PLHIV besides being born in metropolitan or overseas France ? Could the cultural background have an impact on vulnerability and/or vaccine acceptability.

6. Were depression and other psychological aspects considered ? If not, this should be discussed.

7. The discussion of findings in comparison with Bangladesh (line 217) seems less appropriate than comparing with findings from more similar settings in France, Europe. Consider discussing work from the Paris area (Zucman D, et al. The COVID-19 Pandemic and the Migrant Population for HIV Diagnosis and Care Follow-Up: They Are Left Behind. Healthcare. 2022)

8. Introduction : should end with a clear statement of the study objectives, as in the abstract : to explore differences between migrant and non-migrant PLWHIV regarding the socio-economic effects of the COVID-19 crisis and differences in attitudes towards COVID-19 vaccination.

9. Remove “an ad hoc study carried out among four hospitals in Seine- Saint Denis department”

10. Please give the reference for André Grégoire Hospital ethics committee approval

11. “native French” : suggests that the persons had a French background, which is not necessarily the case presumably. Please use “born in France”.

12. Need to define the cohort. “the whole PLWHIV cohort” line 117 and following : reference ? what is whole cohort, how many in the “whole” cohort the 380 patients? Conversely, no difference was observed in terms of age and geographical origin for women between ICOVIH study participants and women from the entire cohort (data not shown).

The manuscript needs careful English language proofreading. For instance :

1. line 56 “where most migrants settle down” does this mean that the department has the highest migrant population in France ?

2. Lines 66-68 : the second sentence belongs in the previous paragraph Attitudes of PLWHIV towards COVID vaccination are also poorly documented. In our respective HIV clinics, we observed an increasing number of migrant patients who had lost their employment, housing, and who were thrown into poverty.

3. Consider rewording “thrown into” which implies a voluntary action

4. line 83 : antiretroviral therapy

5. Covid19 or COVID-19 : use same spelling throughout

6. line 95 : “Annex” is Appendix

7. Key words: SARS-VOC2

8. significatively = significantly

9. lines 133 and below Unemployment prior to the pandemic and unstable jobs (short contract or temporary worker) were more frequent in the migrant group. Being older, French-native participants better to compare group A vs B and not change in the middle pensioner retired

10. Borne = born , kid.s = children

11. Disabled status

12. Native French French-born,

13. line 160-62 not clear, need to correct the English “were reported by non-French-born participant.”

14. “no French-borne” not French-born

15. line 177 for “further hindsight” ??. Self-perception of COVID risk was high in both groups, with only 2.7% of migrants and 3.1% of non-migrants participants considering their good compliance with shielding measures to be sufficient.

16. Non, I am afraid of the side effects but it has nothing to do with my HIV

17. line 199 “born in sub-Saharan Africa”

18. the association has faced a dramatic increase : remove “has”, the statement is in the past tense

19. Line 238 that= then among migrant participants

20. line 246 : “relationship of trust with the doctor who follows up on (would be clearer to say “cares for” … factor “in the use of” could be replace by “for vaccination acceptance.”

21. line 256 : “migrants who are far from” : write “not engaged in” care

22. line 262 and below : “The survey also revealed incidentally” : does not seem incidental.

23. “a highly stressful situation, into no one should be thrown” : needs rewording

Reviewer #2: Review

General comments:

The manuscript addresses an important issue, the impact of social and migratory determinants on the experience of the Covid-19 crisis by people living with HIV in an understudied key French territory. The sample is large with sufficient power for the planned analysis. The main limitation of this work is that the comparison group (non-migrant people living with HIV) is likely to be heterogeneous and belong to specific populations at risk of HIV, therefore not the best comparator for analysing the social determinants of health. However, this does not detract from the interest and originality of this work. The article has value to be accepted after some minor corrections.

Specific comments:

As the majority of the migrants included have arrived in France a long time ago, I suggest that you use the term “immigrant” instead and that you refer to the definition of the French High Council for Integration.

In general, avoid superlatives (much more often -> more often)

Analysis: Why was a multivariate analysis not conducted to assess which social determinants explained the differences observed between people living with HIV from migrant backgrounds and others?

Title: Reword the title: it is not clear whether the "Socio-economic impacts" are those of Covid-19 or vaccination. Avoid using abbreviations in the title

Abstract:

- Write at least one contextualising sentence before the objective

- Detail the abbreviations the first time they appear

- “prior to COVID” epidemic

- Avoid starting your sentences with a number

- Residential insecurity rather than administrative barriers?

- Food instability is not a classical concept: food insecurity or hunger?

- Avoid superlatives (much more often/much higher, etc.)

- Correct “convince the than »

- “thrown into poverty »

Main :

- 44 Define immigrants et migrants terms

- 45 The statement that immigrants are in a disadvantaged social situation needs to be explained by underlining the heterogeneity of this group

- 66: Please formulate the objective of the work more clearly at the end of the introduction

- 70: Please explain in a few words why the Makasi study material is suitable for this study

- 79: We wonder how the doctors found the time to administer the questionnaires in consultation to all the patients, thank you for clarifying this

- 98: please specify the number of the approval of the ethics committee

- 101: please add percentage (participation rate)

Could the number of patients offered the study be compared to the number of consultations of PLWH over the study period in the participating hospitals?

- 101 & 102: please merge the 2 sentences

- 114: Specify the number of first-time migrants (e.g. under 2 years after arrival)

- 116: please add p

- 122: children alone?

- 129: correct born & specify living with HIV

- Table1: Replace legal situation by Administrative status.

How were the patients with a receipt classified?

Please ensure that the conditions for using comparison tests are met for all variables

The p are not always aligned with the variable name, what does this mean?

Some percentages are shifted downwards

- 137: remove the –

- 144: correct: reported by only reported by

- 148: when you test the difference, is it significant?

- 151: were all reported

- 155: avoid superlatives

- 157: what is the difference between “for their doctor’s recommendation or for their doctor » ?

- 158 : correct born/borne

- 167 : barriers measures

- 169 : born

- 173 very/more

- 175 and

- 220: avoid us

As the questionnaire was administered in French for the majority of participants, the French version could also be added as an attachment

Reviewer #3: The paper addresses an important issue and the comparison between migrant and non-migrant populations. A number of areas could be addressed to improve the paper and strengthen it.

Abstract

This needs extending to include aspects of the contribution of the contribution of the paper.

Introduction

This is limited in scope. You need to give a clear account of the research questions and/or hypotheses of the research. You also need some comprehensive backgrounds of the study.

Literature review

You need to consider what has been written on the topic. covid-19 has been going on for sometime and has attracted significant amount of work. You need to reviews the significant literature related to your topic to establish the gap that you are attempting to fill. At present there is no literature review section. This also to reason why your reference list is fairly short with limited significant covid or HIV research.

Methods

The research design needs elaboration. You need a clear justification of your choice of methods. You also need selection criteria of the sample and their characteristics (demographic, professional...). You could comprehensively discuss the data analysis framework at this point.

The region covered is also narrow.

Findings

These present only descriptive statistics in the form of percentages. Some more complex statistics with correlations and factor analysis will strengthen your data analysis.

Discussion needs to integrate the literature. As you have not done a literature review, the references in the discussion are new and the reader cannot appreciate their actual weight in supporting the discussion since they have not been debated before.

You need to state the policy implications as well as the contribution of the paper. You may also discuss the limitations of the research.

Overall the paper needs significant improvements.

6. PLOS authors have the option to publish the peer review history of their article (what does this mean?). If published, this will include your full peer review and any attached files.

Reviewer #1: **Yes: **Professor Laurent Mandelbrot, MD

Reviewer #2: **Yes: **Nicolas Vignier

Reviewer #3: No

---

## [Author Response · Author response to Decision Letter 0]

22 Jun 2023

Reviewer #1: 

1. How did the questionnaire take into account comprehension, language?

Professional translators (« Interservice migrants ») were involved whenever the participant's understanding of French or ability to express him/herself in French had been deemed insufficient. The translator then translated the interviewer's questions and answers, in strict confidentiality of the respondent's identity.

2. How many woman were pregnant and was the proportion different in between groups? pregnancy or planned pregnancy have been shown to …?

Unfortunately, we don't have the end of this question, which was cut off in the e-mail I received from PLOS ONE. Since only one participant was pregnant during the first year of the pandemic (a migrant woman), we can therefore confirm that the proportion of pregnant women did not differ between the two groups. 

3. What was the proportion of drug use in the two groups?

This is a very interesting question, and we have indeed collected data on drug use in the ICOVIH survey. However, due to the wealth of data collected, we decided to analyze the psychological impact and substance use variables separately, and present them in a second step. In relation to our research question: most of the migrants (192/196) never used drugs in our sample, which does not make this consumption a variable of interest for our present study.

4. How many patients with a history of covid in each group?

Adding together suspected and confirmed cases, 10 COVIDs were reported in the French-born group (10.1%) and 27 in the migrant group (13.7%), the difference is not significant. This information has been added to the revised manuscript. 

5. Was there any information on the background of the French-born PLHIV besides being born in metropolitan or overseas France? Could the cultural background have an impact on vulnerability and/or vaccine acceptability?

Unfortunately, we have no information on the cultural background of the participants. In retrospect, we should have collected information on the participants' educational level, which could have served as a first approximation to this background. We have addressed this limitation in a new section entitled "Strengths and Limitations".

6. Were depression and other psychological aspects considered? If not, this should be discussed.

Yes, it's been explored. We have a lot of data on psychological aspects, which will be examined in a separate analysis. Feelings of discouragement and expression of suicidal ideation were equally distributed between migrants and non-migrant participants. 

7. The discussion of findings in comparison with Bangladesh (line 217) seems less appropriate than comparing with findings from more similar settings in France, Europe. Consider discussing work from the Paris area (Zucman D, et al. The COVID-19 Pandemic and the Migrant Population for HIV Diagnosis and Care Follow-Up: They Are Left Behind. Healthcare. 2022)

Absolutely: David Zucman has highlighted the increased risk of follow-up discontinuation among migrant PLWHIV. This reference has been added to justify our concerns about PLWHIV not being receiving timely and equitable access to health care during the crisis. 

8. Introduction: should end with a clear statement of the study objectives, as in the abstract: to explore differences between migrant and non-migrant PLWHIV regarding the socio-economic effects of the COVID-19 crisis and differences in attitudes towards COVID-19 vaccination.

This has been changed in the revised manuscript.

9. Remove “an ad hoc study carried out among four hospitals in Seine- Saint Denis department”

This has been changed in the revised manuscript.

10. Please give the reference for André Grégoire Hospital ethics committee approval

The reference has been added to the revised manuscript.

11. “native French”: suggests that the persons had a French background, which is not necessarily the case presumably. Please use “born in France”.

The terms have been changed in the revised manuscript. 

12. Need to define the cohort. “the whole PLWHIV cohort” line 117 and following: reference? what is whole cohort, how many in the “whole” cohort the 380 patients? 

We have clarified the revised manuscript as follows: the full active file included 1735 PLWHIV of whom 1206 were seen at least once during the study period, and the study was proposed to 380 of them. 

The manuscript needs careful English language proofreading.

The revised manuscript has been fully proof read by a Springer Nature US-trained editor, ensuring the correct use of field-specific terminology in clear and accurate English.

1- line 56 “where most migrants settle down” does this mean that the department has the highest migrant population in France?

Yes. With 30% of its residents foreign-born in 2015 census, Seine Saint Denis is the departement in mainland France with the highest proportion of immigrants. (Insee Analyses Île-de-France n° 114 - Février 2020). The clarification has been made in the revised manuscript.

2. Lines 66-68: the second sentence belongs in the previous paragraph Attitudes of PLWHIV towards COVID vaccination are also poorly documented. In our respective HIV clinics, we observed an increasing number of migrant patients who had lost their employment, housing, and who were thrown into poverty.

Thank you for this remark: this has been changed in the revised manuscript. 

3. Consider rewording “thrown into” which implies a voluntary action

The phrasing has been changed in the revised version

4. line 83: antiretroviral therapy

The wording has been changed as suggested by the reviewer.

5. Covid19 or COVID-19: use same spelling throughout

Wording has been harmonized in the revised manuscript.

6. line 95: “Annex” is Appendix

This has been changed as suggested.

7. Key words: SARS-VOC2

Thank you! This has been corrected to SARS-COV2

8. significantly = significantly

This has been changed as suggested. 

9. lines 133 and below Unemployment prior to the pandemic and unstable jobs (short contract or temporary worker) were more frequent in the migrant group. Being older, French-native participants better to compare group A vs B and not change in the middle pensioner retired

This has been reworded in the revised manuscript. 

10. Borne = born, kid.s = children

11. Disabled status

12. Native French French-born

13 line 160-62 not clear, need to correct the English “were reported by non-French-born participant.”

Changes have been made as suggested and the revised manuscript has been edited by American Journal Experts.

14. “no French-borne” not French-born

We meant: No participant born in France (zero). This has been reworded in the revised manuscript.

15. line 177 for “further hindsight” ??. Self-perception of COVID risk was high in both groups, with only 2.7% of migrants and 3.1% of non-migrants participants considering their good compliance with shielding measures to be sufficient.

This has been reworded in the revised manuscript. 

16. Non, I am afraid of the side effects but it has nothing to do with my HIV

“Non” was corrected to “no” in the revised manuscript.

17. line 199 “born in sub-Saharan Africa”

This has been specified in the revised manuscript. 

18. the association has faced a dramatic increase : remove “has”, the statement is in the past tense

This has been reworded in the revised manuscript. 

19. Line 238 that= then among migrant participants

This has been reworded in the revised manuscript. 

20. line 246 : “relationship of trust with the doctor who follows up on (would be clearer to say “cares for” … factor “in the use of” could be replace by “for vaccination acceptance.”

This has been reworded in the revised manuscript. 

21. line 256 : “migrants who are far from” : write “not engaged in” care

This has been reworded in the revised manuscript. 

22. line 262 and below : “The survey also revealed incidentally” : does not seem incidental.

The term has been removed from the revised manuscript.

23. “a highly stressful situation, into no one should be thrown”: needs rewording

The sentence has been reworded in the revised manuscript. 

Reviewer#2: 

Specific comments:

As the majority of the migrants included have arrived in France a long time ago, I suggest that you use the term “immigrant” instead and that you refer to the definition of the French High Council for Integration.

Migrant has been changed to immigrants and the reference was added to revised manuscript.

In general, avoid superlatives (much more often -> more often)

Superlatives have been removed from the revised manuscript. 

Analysis: Why was a multivariate analysis not conducted to assess which social determinants explained the differences observed between people living with HIV from migrant backgrounds and others?

We thank the reviewer for raising this point: the study was initially designed as a simple comparison between two populations of PLWHIV (immigrants/born in France). As a second step, we considered running a regression model to measure the independent impact of each characteristic on the socio-economic damage each population suffered during the first year of the COVID crisis, but we had chosen a range of variables to explore different socio-economic repercussions and it appeared difficult to reduce them into a single binary variable. Moreover, there were strong correlations between those variables (notably administrative impact) and birth in France versus abroad. However, we suggest in the revised manuscript a logistic regression on vaccine adherence, using the explicative variables available through the questionnaire.

Title: Reword the title: it is not clear whether the "Socio-economic impacts" are those of Covid-19 or vaccination. Avoid using abbreviations in the title

We suggest: Socioeconomic impact of the COVID-19 crisis and early perceptions of COVID-19 vaccines among immigrant and nonimmigrant people living with HIV followed up in public hospitals in Seine-Saint-Denis, France

Abstract:

Write at least one contextualising sentence before the objective

Two sentences of context have been added to the revised abstract. 

Detail the abbreviations the first time they appear

This has been done in the revised abstract. 

“prior to COVID” epidemic

Wording has been changed as suggested. 

Avoid starting your sentences with a number

We have taken this into account in the revised manuscript: sentences do not begin with a number in the revised manuscript. 

Residential insecurity rather than administrative barriers?

Immigrants faced more administrative barriers than French-born patients during the first year of the pandemic (26% versus 9%). They also faced residential insecurity more often that French-born participants (30% versus 7%). These differences are shown in Table 1.

Food instability is not a classical concept: food insecurity or hunger?

The wording has been changed in the revised manuscript.

-Avoid superlatives (much more often/much higher, etc.)

Superlatives were avoided in the revised manuscript.

- Correct “convince the than »

This has been done, thank you.

- “thrown into poverty »

This has been corrected as well.

Main:

- 44 Define immigrants et migrants terms

The term “immigrant” has been defined and the term "migrant" is no longer used.

- 45 The statement that immigrants are in a disadvantaged social situation needs to be explained by underlining the heterogeneity of this group.

This has been done in the revised manuscript. 

66: Please formulate the objective of the work more clearly at the end of the introduction

The double objective was made clear at the end of the revised introduction.

70: Please explain in a few words why the Makasi study material is suitable for this study

In agreement with the main investigator of the MAKASI study, we found it simpler to remove the mention of MAKASI study, as the essential needs we collect in this questionnaire are basic data for all social sciences studies.

79: We wonder how the doctors found the time to administer the questionnaires in consultation to all the patients, thank you for clarifying this

Most doctors have delegated the passing of questionnaires to trained interviewers.

98: please specify the number of the approval of the ethics committee

This has been done in the revised manuscript. 

101: please add percentage (participation rate). Could the number of patients offered the study be compared to the number of consultations of PLWH over the study period in the participating hospitals?

The overall active file was 1735. 1206 PLWHIV were seen at least once during the study period, representing a participation rate of 25%. Due to the short duration of the study and the high workload of the doctors, the study was offered to 380 patients, amongst whom 298 accepted (298/1206=25%). The participation rate has been added at the beginning of the results presentation.

101 & 102: please merge the 2 sentences

These sentences have been completely rewritten in response to previous comments.

Specify the number of first-time migrants (e.g. under 2 years after arrival)

Length of stay was available for 186 of the 197 immigrants surveyed (11 missing data). Only 4 had been living in France for less than 2 years. This clarification has been included in the revised manuscript.

116: please add p

P-value has been added to the revised manuscript.

122: children alone?

Wording has been changed in the revised manuscript.

129: correct born & specify living with HIV

Thank you, this has been done in the revised manuscript.

Table1: Replace legal situation by Administrative status

Wording has been changed in the revised table and in the revised introduction.

How were the patients with a receipt classified?

They were classified as short-residence permit holders.

Please ensure that the conditions for using comparison tests are met for all variables

We have reduced the number of categories describing employment status and household composition, which were previously too numerous, even when using a Fisher exact test. The difference is also significant when we compare the occurrence of any residential difficulty with none.

The p are not always aligned with the variable name, what does this mean?

Some percentages are shifted downwards

We have corrected the misprints in Table 1. The p-value is now aligned with the first modality for categorical variables and with the variable itself for continuous variables.

137: remove the –

144: correct: reported by only reported by

Those changes have been done in the revised manuscript. 

148: when you test the difference, is it significant?

We have reduced the number of categories used to describe the work situation to ensure a robust comparison (cf revised table 1).

157: what is the difference between “for their doctor’s recommendation or for their doctor »?

“…wait for their doctor’s recommendation” refers to the patients who responded “I would only accept if my doctor recommended it.

“…wait for their doctor to convince them” refers to the patients who responded “I could consider vaccination, but only if my doctor convinces me verbally, answering all my questions and explaining why he or she thinks I should be vaccinated.”

151: were all reported

155: avoid superlatives

158 : correct born/borne

167 : barriers measures

169 : born

173 very/more

175 and

220: avoid us

Changes have all been done in the revised manuscript and the manuscript has been edited by American Journal Experts.

As the questionnaire was administered in French for the majority of participants, the French version could also be added as an attachment

The original version has been added as an attachment. 

Reviewer #3: 

Abstract

This needs extending to include aspects of the contribution of the paper.

The contributions of the authors are detailed at the end of the revised abstract.

Introduction

You need to give a clear account of the research questions and/or hypotheses of the research. You also need some comprehensive backgrounds of the study.

The introduction has been rewritten in the light of this comment.

Literature review. You need to consider what has been written on the topic. covid-19 has been going on for sometime and has attracted significant amount of work. You need to reviews the significant literature related to your topic to establish the gap that you are attempting to fill. At present there is no literature review section. This also to reason why your reference list is fairly short with limited significant covid or HIV research.

This has been done in the revised version of the manuscript. 

Methods

The research design needs elaboration. You need a clear justification of your choice of methods. You also need selection criteria of the sample and their characteristics (demographic, professional...). You could comprehensively discuss the data analysis framework at this point.The region covered is also narrow.

Thank you for these comments. We have carried out this study at a regional level, in a pragmatic research perspective. We are a team of clinical doctors, and we worked on the basis of our active files, which are indeed localized, but 60% of sub-Saharan immigrants live in the “Ile de France” (IDF) region, essentially in the north-eastern quarter of the IDF, where our hospitals are located. Our research design intended to be a pragmatic one. We have checked the difference between the PLWHIV we surveyed and the entire active file of our hospitals.

Findings

These present only descriptive statistics in the form of percentages. Some more complex statistics with correlations and factor analysis will strengthen your data analysis.

We ran a logistic multinomial regression model to compare spontaneous vaccine uptake with vaccine refusal on one side, and acceptance based on the physician's recommendation or explanation on the other side. Immigrants and women LWHIV were more likely to accept vaccine on the recommendation of their doctor, or after being convinced by their doctor's explanations, all other things being equal. These analyses show that the trust relationship with the doctor is decisive for women and immigrants living with HIV to accept a new vaccine.

Conversely, PLWHIV who have had an early COVID-19 infection spontaneously seek vaccination, after adjustment for the other characteristics surveyed. This finding contradicts the idea that a patient confident of being immune would be less inclined to take up the vaccine. It suggests, instead, that PLWHIV who had COVID during the first year of the pandemic were generally symptomatic and afraid of getting ill again.

The new analyses are presented and discussed in the revised version of the manuscript.

Conversely, we decided not to run a regression model on the socio-economic impacts of the first year of the pandemic, as the study design - which aimed at a comparison- included too many socio-economic impact outcomes to reduce the model into a single variable.

Discussion needs to integrate the literature. As you have not done a literature review, the references in the discussion are new and the reader cannot appreciate their actual weight in supporting the discussion since they have not been debated before.

Thank you: we took into account this comment in the revised version of the manuscript.

You need to state the policy implications as well as the contribution of the paper. You may also discuss the limitations of the research

Thank you for these suggestions. The public health implications are now addressed at the end of the discussion and in the conclusion. In addition, a paragraph on the study’s limitations has been added to the revised manuscript.

---

## [Decision Letter · Decision Letter 1]

31 Jul 2023

PONE-D-22-26796R1Socioeconomic impact of the COVID-19 crisis and early perceptions of COVID-19 vaccines among immigrant and nonimmigrant people living with HIV followed up in public hospitals in Seine-Saint-Denis, FrancePLOS ONE

Dear Dr. Penot,

Thank you for submitting your manuscript to PLOS ONE. After careful consideration, we feel that it has merit but does not fully meet PLOS ONE’s publication criteria as it currently stands. Therefore, we invite you to submit a revised version of the manuscript that addresses the points raised during the review process.

We look forward to receiving your revised manuscript.

Kind regards,

Ali B. Mahmoud, Ph.D.

Academic Editor

PLOS ONE

Journal Requirements:

Reviewers' comments:

Reviewer's Responses to Questions

**Comments to the Author**

1. If the authors have adequately addressed your comments raised in a previous round of review and you feel that this manuscript is now acceptable for publication, you may indicate that here to bypass the “Comments to the Author” section, enter your conflict of interest statement in the “Confidential to Editor” section, and submit your "Accept" recommendation.

Reviewer #1: (No Response)

Reviewer #2: All comments have been addressed

2. Is the manuscript technically sound, and do the data support the conclusions?

Reviewer #1: Yes

Reviewer #2: Yes

3. Has the statistical analysis been performed appropriately and rigorously? 

Reviewer #1: Yes

Reviewer #2: Yes

4. Have the authors made all data underlying the findings in their manuscript fully available?

Reviewer #1: Yes

Reviewer #2: Yes

5. Is the manuscript presented in an intelligible fashion and written in standard English?

Reviewer #1: Yes

Reviewer #2: Yes

6. Review Comments to the Author

Reviewer #1: The revised manuscript takes into account the reviewers’ suggestion. The English, style and typographic errors have been improved. Most of the questions are addressed, such as the proportions with a history of covid in each group, the administration of questionnaires by trained interviewers, etc. There is a sentence on the study’s weaknesses.

However, a few of the comments still need to be taken into account. They are acknowledged in the response, but no changes are made in the revision. For instance, pregnancy status and drug use, hunger and psychological issues. The reference to Bangladesh was not removed, despite the fact that the issues are quite different from those raised in this study.

The objectives are not stated correctly. The sentence remains informal : “We therefore explored the impact of the COVID-19 pandemic on PLWHIV with a double scope: on the one hand, …and on the other hand...”

Reviewer #2: The authors have satisfacly addressed my comments.

I have only one comment on "Immigrants were 2.4 times more likely to accept [...]" sentence. The odds ratio being ratio of odds, it is not possible to affirm that this multiplies the risk but only that there is a significant association and enriched the manuscript from an analytical and discussion point of view.

The manuscript is in a state of being published

7. PLOS authors have the option to publish the peer review history of their article (what does this mean?). If published, this will include your full peer review and any attached files.

Reviewer #1: **Yes: **Prof Laurent Mandelbrot, MD

Reviewer #2: **Yes: **Nicolas Vignier

---

## [Author Response · Author response to Decision Letter 1]

17 Aug 2023

Reviewer #1: 

The revised manuscript takes into account the reviewers’ suggestion. The English, style and typographic errors have been improved. Most of the questions are addressed, such as the proportions with a history of covid in each group, the administration of questionnaires by trained interviewers, etc. There is a sentence on the study’s weaknesses.

However, a few of the comments still need to be taken into account. They are acknowledged in the response, but no changes are made in the revision. For instance, pregnancy status and drug use, hunger and psychological issues.

Pregnancy status: The pregnancy status of women in both groups is included in the second revision. 

Drug use and psychological issues: Indeed, we had responded to these points in the reply to the reviewer, but had not addressed them in the first revision of the manuscript. We have enhanced the second revision with the data on psychological impact and drug use that we provided in the response to the reviewer's initial comments.

Hunger: We realized that the term we had chosen to describe hunger lacked clarity: we have replaced “food insecurity” with "hunger" in the second revision.

The reference to Bangladesh was not removed, despite the fact that the issues are quite different from those raised in this study.

The reference to Bangladesh has been removed from the second revision. 

The objectives are not stated correctly. The sentence remains informal : “We therefore explored the impact of the COVID-19 pandemic on PLWHIV with a double scope: on the one hand, …and on the other hand...”

The newly revised introduction ends with the formulation of objectives suggested by the Reviewer in his initial recommendations.

Reviewer #2: 

The authors have satisfacly addressed my comments and enriched the manuscript from an analytical and discussion point of view. I have only one comment on "Immigrants were 2.4 times more likely to accept [...]" sentence. The odds ratio being ratio of odds, it is not possible to affirm that this multiplies the risk but only that there is a significant association 

We thank the reviewer for this clarification and modify the second revision of the manuscript accordingly.

---

## [Decision Letter · Decision Letter 2]

8 Oct 2023

Socioeconomic impact of the COVID-19 crisis and early perceptions of COVID-19 vaccines among immigrant and nonimmigrant people living with HIV followed up in public hospitals in Seine-Saint-Denis, France

PONE-D-22-26796R2

Dear Dr. Penot,

We’re pleased to inform you that your manuscript has been judged scientifically suitable for publication and will be formally accepted for publication once it meets all outstanding technical requirements.

Kind regards,

Ali B. Mahmoud, Ph.D.

Academic Editor

PLOS ONE

Additional Editor Comments (optional):

Reviewers' comments:

Reviewer's Responses to Questions

**Comments to the Author**

1. If the authors have adequately addressed your comments raised in a previous round of review and you feel that this manuscript is now acceptable for publication, you may indicate that here to bypass the “Comments to the Author” section, enter your conflict of interest statement in the “Confidential to Editor” section, and submit your "Accept" recommendation.

Reviewer #1: All comments have been addressed

Reviewer #2: All comments have been addressed

2. Is the manuscript technically sound, and do the data support the conclusions?

Reviewer #1: Yes

Reviewer #2: Yes

3. Has the statistical analysis been performed appropriately and rigorously? 

Reviewer #1: Yes

Reviewer #2: Yes

4. Have the authors made all data underlying the findings in their manuscript fully available?

Reviewer #1: Yes

Reviewer #2: Yes

5. Is the manuscript presented in an intelligible fashion and written in standard English?

Reviewer #1: Yes

Reviewer #2: Yes

6. Review Comments to the Author

Reviewer #1: (No Response)

Reviewer #2: The changes have been made in accordance with my comments. The manuscript is then ready for publication.

7. PLOS authors have the option to publish the peer review history of their article (what does this mean?). If published, this will include your full peer review and any attached files.

Reviewer #1: **Yes: **Prof Laurent Mandelbrot

Reviewer #2: **Yes: **Nicolas Vignier

---

## [Editor Report · Acceptance letter]

13 Oct 2023

PONE-D-22-26796R2 

Socioeconomic impact of the COVID-19 crisis and early perceptions of COVID-19 vaccines among immigrant and nonimmigrant people living with HIV followed up in public hospitals in Seine-Saint-Denis, France 

Dear Dr. Penot:

I'm pleased to inform you that your manuscript has been deemed suitable for publication in PLOS ONE. Congratulations! Your manuscript is now with our production department. 

Kind regards, 

on behalf of

Dr. Ali B. Mahmoud 

Academic Editor

PLOS ONE